# Antibody Responses to SARS-CoV-2 Infection—Comparative Determination of Seroprevalence in Two High-Throughput Assays versus a Sensitive Spike Protein ELISA

**DOI:** 10.3390/vaccines9111310

**Published:** 2021-11-11

**Authors:** Dinesh Mohanraj, Kelly Bicknell, Malini Bhole, Caroline Webber, Lorna Taylor, Alison Whitelegg

**Affiliations:** 1Faculty of Medicine, Biology and Health, The University of Manchester, Manchester M13 9PL, UK; 2Department of Medical Microbiology, Portsmouth Hospital University NHS Trust, Portsmouth PO6 3LY, UK; Kelly.bicknell@porthosp.nhs.uk; 3Department of Clinical Immunology, The Dudley Group NHS Foundation Trust and Black Country Pathology Service (BCPS), West Midlands DY1 2HQ, UK; malini.bhole@nhs.net; 4Department of Clinical Immunology, Black Country Pathology Service (BCPS), West Midlands DY1 2HQ, UK; caroline.webber@nhs.net (C.W.); Lorna.Taylor1@nhs.net (L.T.); 5Department of Blood Sciences, Portsmouth Hospital University NHS Trust, Portsmouth PO6 3LY, UK; Alison.whitelegg@porthosp.nhs.uk

**Keywords:** SARS-CoV-2, Roche Elecsys SARS-CoV-2 antibody assay, Abbott IgG antibody assay, The Binding Site total antibody assay

## Abstract

Robust assay development for SARS-CoV-2 serological testing requires assessment of asymptomatic and non-hospitalised individuals to determine if assays are sensitive to mild antibody responses. Our study evaluated the performance characteristics of two high-throughput SARS-CoV-2 IgG nucleocapsid assays (Abbott Architect and Roche) and The Binding Site (TBS) Anti-Spike IgG/A/M ELISA kit in samples from healthcare workers (HCWs). The 252 samples were collected from multi-site NHS trusts and analysed for SARS-CoV-2 serology. Assay performance was evaluated between these three platforms and ROC curves were used to redefine the Abbott threshold. Concordance between Abbott and TBS was 66%. Any discrepant results were analysed using Roche, which showed 100% concordance with TBS. Analysis conducted in HCWs within 58 days post-PCR result demonstrated 100% sensitivity for both Abbott and Roche. Longitudinal analysis for >100 days post-PCR led to sensitivity of 77.2% and 100% for Abbott and Roche, respectively. A redefined Abbott threshold (0.64) increased sensitivity to 90%, producing results comparable to TBS and Roche. The manufacturer’s threshold set by Abbott contributes to lower sensitivity and elevated false-negative occurrences. Abbott performance improved upon re-optimisation of the cut-off threshold. Our findings provided evidence that TBS can be used as bespoke alternative for SARS-CoV-2 serology analysis where high-throughput platforms are not feasible on site.

## 1. Introduction

Severe acute respiratory syndrome coronavirus 2 (SARS-CoV-2), the virus responsible for coronavirus disease 2019 (COVID-19), has led to a global pandemic with more than 183 million confirmed infections and 3.9 million fatalities [1]. Since the initial identification of COVID-19, mass testing and case determination have been a mainstay strategy for controlling viral transmission and guiding both the public health and political strategies required to mitigate the deleterious consequences of SARS-CoV-2. 

SARS-CoV-2 testing is represented by the initial detection of the virus in nasopharyngeal specimens by real-time polymerase chain reaction (RT-PCR), which is currently considered as the gold standard for confirming suspected diagnosis and identifying asymptomatic carriers [2]. Complementing RT-PCR investigation is serological testing, which utilises immunoassays for the detection of SARS-CoV-2 antibodies as a measure of the adaptive immune response to natural infection and/or vaccines. Immunoassays either detect specific antibody types (such as immunoglobulin M or immunoglobulin G) or the total antibody. However, antibody detection usually occurs 5–7 days post-infection; thus, it is not representative of acute infection [3]. Whilst the degree and duration of immunity conferred by these antibodies are currently unclear, the widescale use of serological testing has been applied at the population level to establish population exposure. Other uses of serological analysis include assessing the individual infection risk and measuring humoral immunity elicited in vaccine trials [4]. 

In response, several immunoassays have been designed by manufacturers to meet global laboratory infrastructures, enabling high-throughput serological analysis. Two of the earliest tests to become commercially available both measured IgG antibodies against the SARS-CoV-2 nucleocapsid protein. The Abbott SARS-CoV-2 IgG assay was initially validated on sera taken from hospitalised COVID-19 patients at ≥14 days post RT-PCR (n = 31) and 997 pre-pandemic sera, and reported a sensitivity and specificity of 100% (95% CI: 95.89–100.00) and 99.6% (95% CI: 98.98–99.89), respectively [5]. The Roche Elecsys Anti-SARS-CoV-2 assay was initially validated using sera from COVID-19 patients at ≥14 days post RT-PCR (n = 102) and 10,453 pre-pandemic sera, and reported a sensitivity and specificity of 99.5% (95% CI: 97.0–100%) and 99.80% (95% CI: 99.69–99.88%), respectively [6]. 

The majority of reports on the performance of serological assays have been performed on samples from hospitalised patients with severe COVID-19, who have a high viral load and elicit robust immune responses, which may in part account for the high sensitivities reported. To date, there are limited studies that have focused on evaluating assay performance in non-hospitalised or community-based COVID-19 cases. From the limited data available in such patients, studies have demonstrated the poor sensitivity of the Abbott assay in the detection of low levels of antibodies. For instance, a pre-print report highlighted a sensitivity of just 61.5% when investigating community-based COVID-19 cases (n = 26) [7]. 

Understanding antibody responses in asymptomatic and non-hospitalised individuals is of major importance for ascertaining viral transmission and for SARS-CoV-2 serological assay development [8]. As a result, it is crucial that antibody assays are able to detect low levels of antibodies in order to be of use in both clinical and sero-epidemiological settings. The Binding Site (TBS) Anti-IgG/A/M SARS-CoV-2 ELISA (TBS, Birmingham, UK) measures IgG, IgA, and IgM antibodies against the SARS-CoV-2 trimeric spike glycoprotein, and has been shown to detect antibodies in PCR-confirmed non-hospitalised asymptomatic COVID-19 patients and patients with mild disease [9]. 

To date, limited direct assessments of multiple immunoassays have been conducted on large data sets from non-hospitalised patients. We therefore sought to evaluate the Abbott, Roche, and TBS immunoassays in samples taken from healthcare workers to identify assay sensitivities and redefine the assay thresholds required for optimisation.

## 2. Methods

### 2.1. Patients and Study Design

The urgent need to better understand SARS-CoV-2 seroprevalence during the first wave of the pandemic led to regulatory bodies, such as the Food and Drug Administration (FDA) and the Medicines and Healthcare products Regulatory agency (MHRA), to approve the Abbott Architect nucleocapsid (NC) IgG assay (hereon referred to as Abbott) [Abbott, Chicago, IL, USA] and Elecsys^®^ Anti-SARS-CoV-2 IgG nucleocapsid immunoassays [Roche, Basel, Switzerland]. In the UK, these were rolled out as generic assays, as part of a wider national screening programme to determine seroprevalence in healthcare workers and to determine prior SARS-CoV-2 exposure in clinical settings. The mandatory and fast-tracked deployment of such assays meant that health institutions and requestors did not have access to extensive assay performance characteristics to make informed choices. 

This study was conducted at a time when no licensed SARS-CoV-2 vaccines were available; therefore, positive antibody responses correlated with natural infection. It is well known that following SARS-CoV-2 infection, humoral responses elicit an early response against both the NC and spike proteins [10]. Such a polyclonal response demonstrates a flexibility of antigen choice, spike or NC, in immunoassays, as natural infection should stimulate the humoral response to multiple SARS-CoV-2 structural sites. Therefore, we compared and evaluated serological outcomes between two high-throughput immunoassays, Abbott and Roche, alongside The Binding Site (TBS) human anti-spike IgG/A/M SARS-CoV-2 ELISA [The Binding Site, Birmingham, UK]. 

A comparative evaluation of the above immunoassays was conducted at two different sites: Queen Alexandra Hospital, Portsmouth Hospitals University NHS Trust (PHU), and the Dudley Group of Hospitals NHS Foundation Trust (DGH). At PHU, this study was approved by Portsmouth Hospital NHS Trust Research Ethics Committee. Serum samples (n = 188) were collected that had been provided as part of the SIREN research study (Clinicaltrials.gov identifier: NCT12345678). Healthcare workers aged 18 years or older provided fortnightly blood samples for SARS-CoV-2 serology surveillance. Only five participants had a positive SARS-CoV-2 RT-PCR prior to SIREN enrolment. At DGH, serum from healthcare workers (n = 64), aged 18 years or older, was collected; 39 samples were confirmed RT-PCR-positive for SARS-CoV-2 and 9 were RT-PCR-negative and symptom-negative for SARS-CoV-2 [11]. 

### 2.2. Anti-SARS-CoV-2 Assays

Samples from PHU were initially analysed for SARS-CoV-2 antibodies at the Department of Medical Microbiology, PHU, using the Abbott nucleocapsid IgG assay [Abbott, Chicago, IL, USA] as part of the SIREN study. Samples were identified as positive or negative based on the manufacturer’s cut-off (Appendix A). Samples were further evaluated using The Binding Site (TBS) human anti-IgG/A/M SARS-CoV-2 ELISA [The Binding Site, Birmingham, UK]. If discordance was reported between the TBS and Abbott platforms, the samples were evaluated using the Elecsys^®^ Anti-SARS-CoV-2 IgG nucleocapsid immunoassay [Roche, Basel, Switzerland] at Poole NHS Trust. Samples from DGH were analysed using the Abbott nucleocapsid IgG assay [Abbott, Chicago, IL, USA] and further evaluated using the Elecsys Anti-SARS-CoV-2 IgG nucleocapsid immunoassay [Roche, Basel, Switzerland]. Assays were conducted in accordance with the manufacturers’ standard operating procedures by HCPC-registered laboratory staff in laboratories that held United Kingdom Accreditation Service accreditation. All assays were conducted with specified controls and calibrants using clinical cut-off thresholds for negative and positive as determined by the manufacturers. 

### 2.3. Procedures

Primary serum separator tubes were stored at 2–8 °C for up to seven days after venepuncture. Selected samples for each run were derived within this seven day period and primary samples were centrifuged for 10 min at 3500× *g*. Supernatant (0.5–1 mL) was derived and made up into sample aliquots, which were stored at −20 °C until they were processed by the respective immunoassay. Samples were limited to fewer than three freeze-thaw cycles. 

### 2.4. Statistical Analysis

Analyses were performed using Graphpad Prism v9.0.1 (La Jolla, CA, USA) and SPSS v27.0 (IBM Corp, Armonk, NY, USA). Fisher’s exact test was used to evaluate categorical data between immunoassays, whereas Chi-squared analysis was used to evaluate categorical data as a three-way comparison method. One-sample *t*-test was employed to determine the significance of result outcomes produced by each immunoassay. Sensitivity and specificity with exact binomial 95% CI for assays was carried out by Fisher’s exact test. Assessment of agreement between immunoassays was conducted by concordance (percentage) and by ƙ-Cohen. Receiver operator characteristic (ROC) curves defined trade-offs in assay sensitivity and specificity for Abbott immunoassay. 

### 2.5. Ethical Considerations

This study was conducted based on service evaluation, which did not require ethical approval. Samples were derived from altruistic healthcare volunteers and were designed for assay verification. All samples were derived after informed consent was gained from volunteers.

## 3. Results

### 3.1. PHU Study: Comparison between Abbott and TBS

Comparing antibody levels using the Abbott and TBS assays in 188 serum samples taken from healthcare workers (HCWs) found concordant results in 125 samples (66.4%, Figure 1A), demonstrating a fair agreement between assays as measured using ƙ-Cohen (ƙ = 0.377, 95% CI: 0.27–0.48, SE = 0.054). A two-fold decrease in positive antibody detection was seen with the Abbott assay compared to TBS, demonstrating a significant difference in outcomes between assays (Fisher’s exact test, *p* < 0.0001). Of the samples reported as negative using the Abbott assay (n = 131), 59 were reported as positive using the TBS assay (*p* < 0.0001, Figure 1B). As seen in Figure 1B, a clear demarcation of positive and negative samples can be seen using the TBS assay. In these 59 samples, the results from the Abbott assays fell within the index values of 0.26–1.31 (Figure 1C). 

Of these 59 discordant samples between Abbott and TBS analysis, 48 samples were sent to Poole NHS Trust for analysis using the Roche Elecsys Anti-SARS-CoV-2 IgG nucleocapsid assay. Only one sample result (1/48) was in concordance between Roche and Abbott, demonstrating poor agreement (ƙ-Cohen= −0.133, 95% CI: −0.26–0.02). whereas all 48 samples analysed using Roche produced outcomes that were 100% in agreement with results generated from TBS. Comparative analysis of 48 samples across the three immunoassays demonstrated that Abbott reported negative outcomes in 45 samples, whereas negative outcomes were only observed in 4 samples from TBS and Roche. Overall, a significant difference in outcomes was generated by Abbott in comparison to Roche and TBS immunoassays (Chi-square, *p* < 0.0001, Figure 2).

### 3.2. DGH Study: Evaluation of Sensitivity and Specificity between Roche and Abbott

Serum samples were collected from 48 healthcare workers (HCWs) within 58 days of registering a RT-PCR result or the onset of COVID-19 symptoms (PCR positive: 39, PCR negative: 9). Further, 16 samples were collected from HCWs where RT-PCR analysis was not performed and showed no signs and symptoms of COVID-19 infection; these were classified as presumed COVID-19 negatives. Amongst these samples, 28 were followed for >100 days post-PCR result or symptom onset (PCR-positive: 22, PCR-negative: 6). 

In samples analysed within ≥19–≤58 days, the Abbott and Roche assays showed 100% concordance and perfect agreement (ƙ-Cohen: 1.000). All assay characteristics for both the Roche and Abbott assays were identical (Table 1); both assays reported sensitivity and specificity of 97.4% and 80%, respectively, with an LR of 4.87. 

Samples analysed after >100 days showed a concordance of 79% (22/28) with a moderate agreement (ƙ-Cohen: 0.462, 95% CI: 0.15–0.78). Of the six discordant samples, which were reported as negative using the Abbott assay, was shown to be positive using the Roche assay. Of these, 5/6 were PCR-positive at initial testing, and the sample that was PCR-negative tested positive in both assays at the earlier timepoint. 

### 3.3. DGH Abbott–Roche Comparison Consistent with PHU Study Discordant Range

To further interrogate the assay performance, the samples for which low positive results were recorded (n = 20, range: 1.28–4.00 index value) using the Roche assay were compared to the result generated using the Abbott assay (Figure 3). Concordance was found in only 4/20 samples (20%). Consistent with the findings from PHU, most discrepancies arose in samples that had a result between 0.26–1.31 according to the Abbott assay. This further exemplified the inferior sensitivity of the Abbott assay, which gave significantly more false negatives (Fisher’s exact test, *p* < 0.0001) in comparison to the Roche assay. 

### 3.4. Definition of Thresholds Harmonising Abbott, TBS, and Roche Results

The above analyses allowed us to define the optimal threshold enabling concordance and agreement between the three assays (Figure 4). Based on the ROC analysis, a new threshold of ≥0.64 for the index value of the Abbott assay was selected. Using the ROC curve, this threshold gave a sensitivity of 90.91% (95% CI: 70.84–98.88%), a specificity of 83.33% (95% CI: 35.88–99.58%), and an LR of 5.45, whereas the manufacturer’s threshold of 1.4 gave a sensitivity of 77.27% (95% CI: 54.63–92.18%), a specificity of 83.3% (95% CI: 35.88–99.58), and an LR of 4.64. 

Implementation of this redefined threshold to data obtained at PHU (n = 188 samples) gave a concordance of 82% (155/188) between the Abbott and TBS assays, where outcomes between assays were not significantly different (Fisher’s exact test, *p* = 0.0621, Figure 5A). This represented an increase in concordance of 16% and resulted in good agreement (ƙ-Cohen: 0.650, 95% CI: 0.54–0.76, SE: 0.054). 

No significant difference in results between the Roche and Abbott assays was established when the redefined threshold was applied to samples analysed at >100 days post-PCR result (Fisher’s exact test, *p* = 0.7458, Figure 5B). Concordance between assays was 89%, which represented an increase of 10% compared to the manufacturer’s threshold, and agreement was good between both assays (ƙ-Cohen: 0.650, 95% CI: 0.33–1.00, SE: 0.054). Concordance in low positive Roche samples (n = 20) was improved from 20% to 65%. Together, these findings demonstrate that the use of the redefined threshold enhanced the sensitivity of the Abbott assay in line with that reported by both the Roche and TBS assays. 

## 4. Discussion

SARS-CoV-2 serological assays have a pivotal role in assessing community transmission, especially in the evaluation of seroprevalence of both asymptomatic and symptomatic cases. Based on several meta-analyses and epidemiological baseline modelling, around 30% of infected individuals display no symptoms [12], signifying a major source of asymptomatic transmission. Thus, SARS-CoV-2 antibody tests could be used to define an accurate extent of an outbreak and highlight its geographical distribution and hotspots. To achieve this, serological assays need to confer enhanced sensitivity, where antibodies against the SARS-CoV-2 nucleocapsid and Spike proteins can be detected, even at lower levels. Abbott SARS-CoV-2 IgG immunoassay detects NC antibodies with claimed sensitivities of 98.3–100% [13,14,15,16]. However, the majority of these claims have been obtained from samples derived from patients hospitalised with severe COVID-19, who have high viral loads and robust humoral responses. Here, we demonstrated that sensitivity of Abbott was overestimated using the current manufacturer’s thresholds in a comparative evaluation performed alongside TBS and Roche assays. 

We collected serum samples from healthcare workers (HCWs) at two different sites and evaluated the performance, with a focus on assay sensitivities, conferred by the different SARS-CoV-2 antibody assays: TBS, Abbott and Roche. Samples from HCWs were stratified according to the data available at each site. At Portsmouth University Hospital (PHU), samples were stratified into positive and negative as reported by the Abbott assay, and were later evaluated using the TBS assay. As samples from HCWs were collected during the first wave, mass RT-PCR testing was not underway; thus, RT-PCR data were not available for most of these participants. Samples collected at DGH were stratified into COVID-19-positive and -negative samples according to RT-PCR. At both sites, concordance and agreement of SARS-CoV-2 serology were compared between the different immunoassays. 

Significant discrepancies were identified between the Abbott and TBS assays, with only 66% concordance found. Analysis of discrepant samples using the Roche assay showed perfect agreement with the TBS assay, whilst only one result was concordant between the Roche and Abbott assays. This highlighted the potential for the significant underestimation of seroprevalence using the Abbott assay. Of note, the sensitivity and specificity (97.4% and 80.0%, respectively) of the Abbott and Roche assays were identical in samples tested at ≤58 days post-PCR or symptom onset, but differed significantly in samples tested at >100 days post-PCR or symptom onset (77.3% by Abbot compared to 100% by Roche). As SARS-CoV-2 antibody levels naturally wane over time [17], this would suggest that the Abbott assay lacks the sensitivity to detect lower antibody levels, which can be detected using the Roche and TBS assays. 

We hypothesised that the reasoning behind the lower sensitivity seen in Abbott was primarily influenced by the high cut-off threshold set by the manufacturer. Such speculation was supported by our findings, as using ROC analysis we defined a cut-off of 0.64, which gave a sensitivity of 90.0% and showed good agreement with the Roche and TBS assay. Whilst it does not meet the MHRA assay stipulation of 98% sensitivity and specificity [18], the optimisation of this redefined threshold led to fewer false-negative results. 

Our findings are supported by a recent study that used anosmia and ageusia reported by HCWs as a high pre-test probability of mild and asymptomatic COVID-19 infections [17]. Correlation was observed between a high proportion of these reported anosmia/ageusia, and increased antibody readings below the manufacturer’s thresholds. Adjustment for reported anosmia and ageusia produced an estimated test sensitivity for the Abbott of 79.3%. It should be noted that this referred study uses self-reported symptoms of anosmia and ageusia, which may be subjective. Furthermore, given the lack of a pre-COVID-19 group exposed to these same questions, there is some uncertainty in attributing these symptoms uniquely to SARS-CoV-2 and not other respiratory viral illness. 

Furthermore, the low sensitivity of the Abbott assay was also demonstrated in results from the Spanish national serologic survey [19], which tested 5118 individuals with typical COVID-19 symptoms within 14 days of symptom onset. It was found that only 18.0% of these individuals had SARS-CoV-2 antibodies. The authors interpreted this as a “sizable proportion of suspected cases might not have been caused by SARS-CoV-2”. We find this explanation implausible and suggest that this data further supports our consensus that the Abbott assay underestimates SARS-CoV-2 seroprevalence with the use of the current manufacturer’s threshold. It is our opinion that the Abbott assay used in the Spanish study conferred low sensitivity, which is supported by our findings of 77% sensitivity from our data derived from two different hospital sites. 

Another explanation for the seroprevalence underestimation observed in this Spanish population could be due to the sampling time used in the study (14 days post-symptom onset). Given the dynamics of humoral responses, 14 days post-viral challenge may not be sufficient time to prompt an effective adaptive response. Thus, we recommend that any future sero-epidemiological studies choose a sampling point post-symptom or PCR result that enables apt time for the development of detectable antibodies. 

Whilst sensitivity and specificity are crucial parameters for any serological assay, estimating the PPV and NPV is useful when determining clinical and sero-epidemiological utility. Our data demonstrate that Abbott’s low NPV (0.5) is not sufficient to rule out an individual’s recent exposure to SARS-CoV-2, which calls into question the utility of the assay when determining seroprevalence. As this study demonstrated, the TBS assay demonstrated enhanced sensitivity, as it was able to detect low-level antibody responses. This was supported by a recent study demonstrating that the TBS assay was able to detect antibody responses in non-hospitalised asymptomatic COVID-19 patients and patients with mild disease [9]. However, a significant proportion of these low-level antibody responses was not detected by Abbott assay when using the manufacturer’s cut-off threshold. However, re-optimisation of a new cut-off level (0.64) significantly improved the sensitivity of Abbott for detecting low-level antibody responses, which was comparable to both TBS and Roche. 

The current licenced COVID-19 vaccines target the Spike protein of SARS-CoV-2, consequently eliciting humoral responses against the Spike protein. However, Roche and Abbott assays both measure antibodies against the SARS-CoV-2 nucleocapsid protein; thus, they may not be of use in determining vaccination responses. Uniquely, the TBS assay detects SARS-CoV-2 antibodies against the trimeric spike protein, which is the predominant immune response produced in response to SARS-CoV-2 vaccination. It is our belief that the longitudinal measurement of antibody responses following vaccination should be conducted using multiple assays that measure antibodies against the spike protein, and that this should be done alongside frequent RT-PCR testing. Such experimental design may allow the determination of protective antibody levels, as well as cross-comparative evaluation of the performance of several anti-spike serological assays. We envisage this as a multi-national study, analogous to the current SIREN study. Such studies are urgently required. 

Nevertheless, we believe nucleocapsid assays still have a role in sero-epidemiological surveillance. We envisage dual serological testing from use of nucleocapsid and spike assays. This will enable the identification of new seroconverts from natural infections and the monitoring of vaccination responses in niche clinical scenarios, such as in poor vaccine responders or the immunocompromised. It is paramount that proficient clinical interpretation is available in sites utilising such diagnostic strategies. Moreover, we have provided evidence here of the enhanced sensitivity of the TBS assay, which can be utilized as a viable alternative that meets bespoke clinical and possible geographical needs where high-throughput assays are unavailable on site.

The present study has some limitations. Firstly, our sample sets may under-represent ethnic groups, did not comprise children, and did not capture sufficient clinical metadata required for full clinical and analytical result interpretation. We propose a further study to be conducted, where, along with the participant’s PCR status and clinical symptoms, the investigation of these immunoassays to detect responses in asymptomatic and mild cases should be evaluated to deduce the clinical sensitivity of different immunoassays. Whilst limited, the study highlights that the TBS assay showed enhanced sensitivity in detecting mild and asymptomatic antibody responses [9]. 

## 5. Conclusions

We demonstrate here that the Abbott manufacturer’s cut-off threshold leads to a significant increase in false-negative outcomes, which contributes to the underestimation of seroprevalence. The manufacturer’s cut-off at 1.4 overexaggerates the dynamics of antibody waning, as seropositivity transitions into seronegative results much more quickly in participants analysed by Abbott in comparison to Roche and TBS. Our findings provide evidence that the Abbott manufacturer’s threshold should be redefined (0.64 index value), which increases sensitivity (90%) and reduces the incidence of false negatives.

## Figures and Tables

**Figure 1 vaccines-09-01310-f001:**
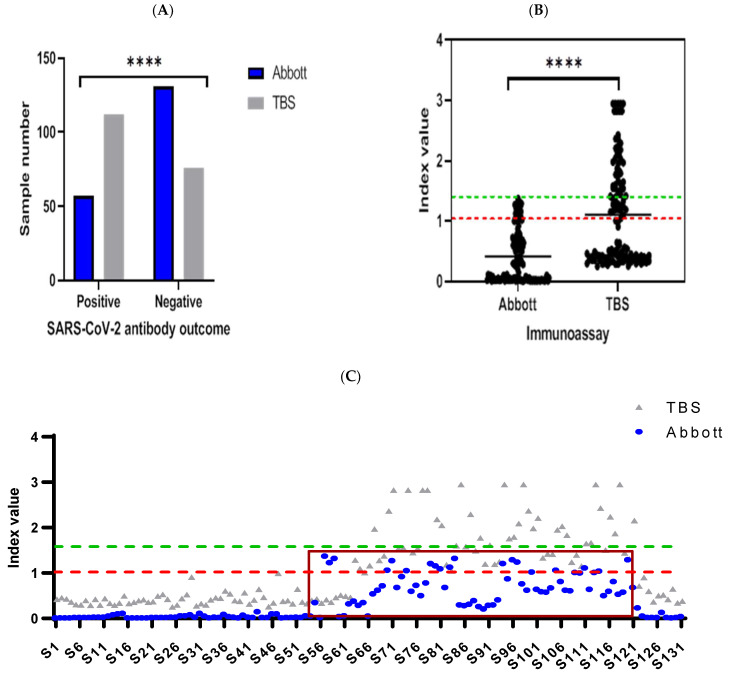
Comparative evaluation between Abbott and TBS, which detect antibody responses against SARS-CoV-2 nucleocapsid and spike, respectively. Results are presented as (**A**) bar graph, which compares positive and negative SARS-CoV-2 antibody outcomes in 188 samples analysed between Abbott and TBS. Statistically significant differences for positive versus negative outcomes derived from both assays (**** = *p* < 0.0001, Fisher’s exact test). Nested scatterplot is presented by (**B**), which portrays the difference in outcomes produced by TBS in analysis of 131 samples deemed negative by Abbott (green dashed line = cut-off for Abbott (1.40 index value)). Of TBS outcomes, 45% were positive (red dashed line = cut-off for TBS (1.0 index value)) and above TBS mean index value (1.06, 95% CI: 0.92–1.20), which was significantly higher compared to TBS (0.36 index value, 95% CI: 0.31–0.46, **** = *p* < 0.0001, one sample *t*-test). Paired index values derived from negative sample analysis, from Abbott and TBS, are presented on the scatterplot in (**C**). Discrepant results from Abbott (blue dots) are categorised within the brown rectangular box (n = 59, index values; 0.26–1.31), where reciprocal analysis by TBS (grey triangles) was deemed positive in all 59 samples. The x-axis represents sample identity (S1 = sample 1) derived from individuals for analysis in Abbott and TBS.

**Figure 2 vaccines-09-01310-f002:**
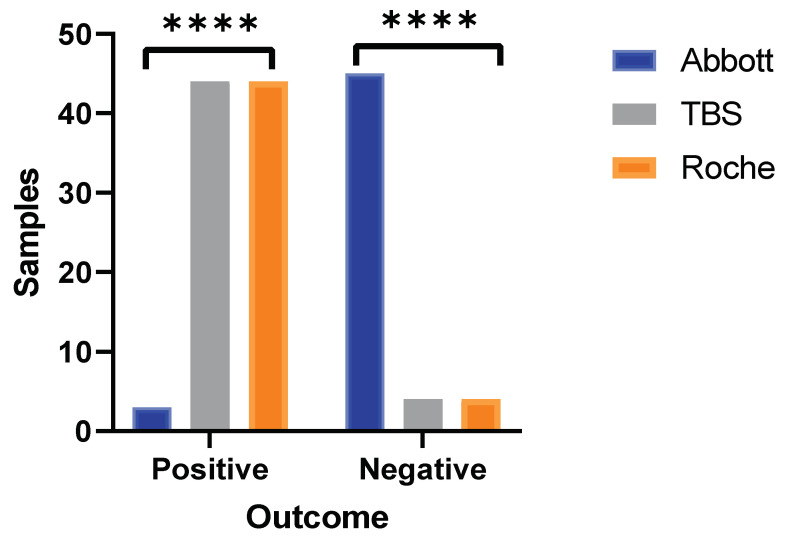
Three-way method comparison between Abbott, TBS, and Roche. Bar graph portrays SARS-CoV-2 antibody outcomes produced by Roche in the discordant sample set (n = 48) between Abbott and TBS. Statistically significant differences observed from outcomes generated by Abbott versus Roche and TBS (**** = *p* < 0.0001, Chi-square test). Outcomes produced by both TBS and Roche are identical to each other (100% concordance).

**Figure 3 vaccines-09-01310-f003:**
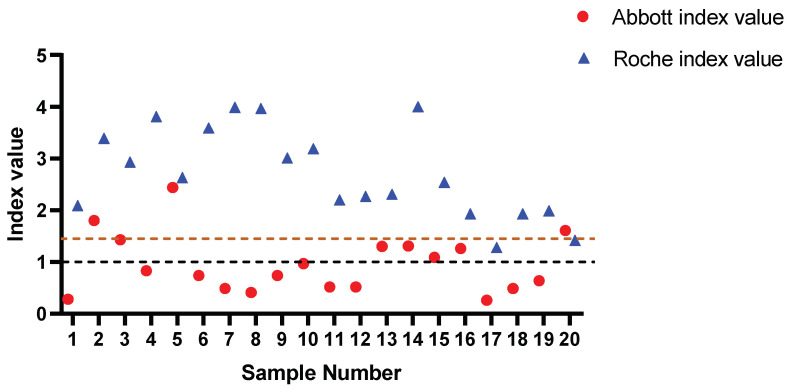
Roche-reported low-level positive samples analysed by Abbott. Statistically significant difference in reported outcomes observed in low-level positive samples analysed by Abbott. All discordant outcomes (n = 16) reported by Abbott were below the manufacturer’s set threshold (brown dashed line) and comprised index values of 0.26–1.31, consistent with PHU Abbott versus TBS discordant range. Black dashed line = Roche manufacturer threshold. Brown dashed line = Abbott manufacture threshold.

**Figure 4 vaccines-09-01310-f004:**
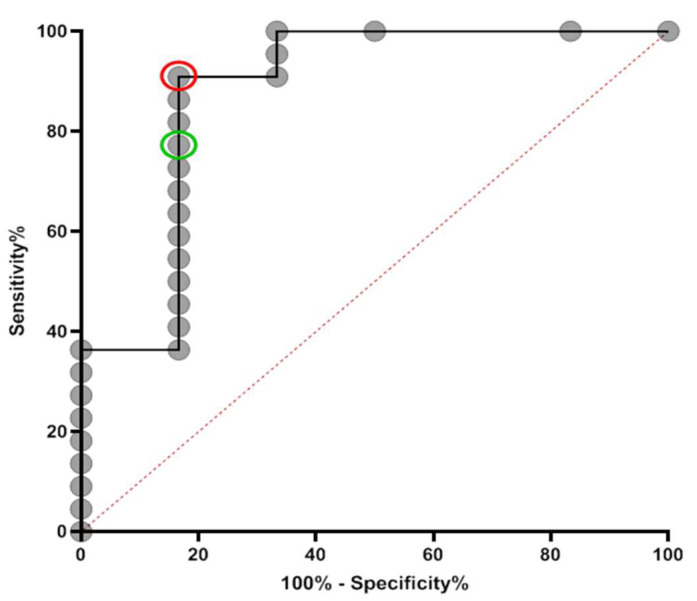
ROC curve for Abbott at specification of samples taken >100 days post-PCR result. The new cut-off (sensitivity = 90.91% (95% CI: 70.84–98.88%), specificity = 83.33% (95% CI: 35.88–99.58%)) and manufacturer’s cut-off (sensitivity = 77.27% (95% CI: 54.63–92.18%), specificity = 83.3% (95% CI: 35.88–99.58)) are represented by red and green circles, respectively. Data are presented for 22 RT-PCR-positive samples and 6 known RT-PCR-negative samples. Area under curve = 0.878, SE: 0.0991, 95% CI: 0.6845 to 1.000.

**Figure 5 vaccines-09-01310-f005:**
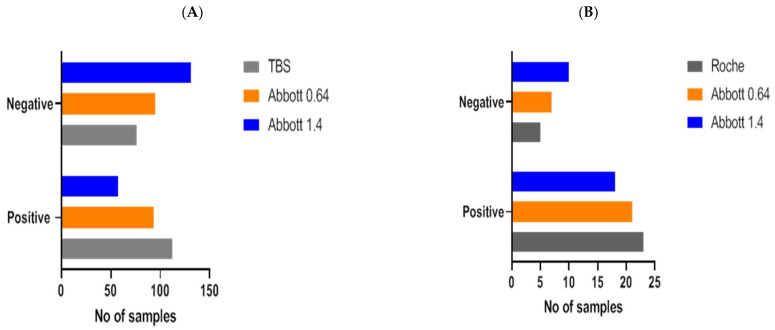
SARS-CoV-2 antibody outcomes (positive, negative) were compared with our redefined Abbott threshold (Abbott 0.64) and the manufacturer’s threshold (Abbott 1.4) in samples analysed at PHU (**A**) and DGH (**B**). Serological outcomes derived from redefined and manufacturer’s Abbott threshold were also compared with outcomes reported by TBS (**A**) and Roche (**B**). In comparison to the Abbott manufacturer’s threshold, the use of Abbott 0.64 demonstrated no significant differences and improved concordance and agreement between serological outcomes produced by TBS and Roche.

**Table 1 vaccines-09-01310-t001:** Assay performance characteristics of Roche and Abbott in analysis of HCWs samples both within 58 days and >100 days of PCR result or symptom onset. Specificity was assessed using PCR-negative or no-onset of COVID-19 symptoms samples and sensitivity was assessed using RT-PCR positive samples. CI = Confidence interval. SE = Standard error.

Parameter	Within 58 Days Post-PCR/Symptom Onset		>100 Days Post-PCR/Symptom Onset	
	Abbott	Roche	Abbott	Roche
Sensitivity	97.4% (95% CI: 86.8–99.8%)		77.3% (95% CI: 56.6–89.9%)	100% (95% CI: 85.1–100%)
Specificity	80.0% (95% CI: 60.9– 91.1%)		83.3% (95% CI: 43.7–99.2%)	
PPV	0.88 (95% CI: 0.75–0.95)		0.94 (95% CI: 0.74–0.99)	0.96 (95% CI: 0.79–0.99)
NPV	0.95 (95% CI: 0.77–0.99)		0.50 (95% CI: 0.24–0.76)	1.00 (0.57–1.00)
LR	4.87		4.64	6.00
ƙ-Cohen	1.000 SE: 0.000		0.462 SE: 0.164	

## Data Availability

Data available upon request.

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
