# Peer review of "Antibody Responses to SARS-CoV-2 Infection—Comparative Determination of Seroprevalence in Two High-Throughput Assays versus a Sensitive Spike Protein ELISA"

_vaccines, 2021, doi:10.3390/vaccines9111310_

Round 1

Reviewer 1 Report

The work of Mohanraj and colleagues compares the ABBOTT ROCHE and TBS tests for the assessment of antibody response for SARS-CoV2.
The work of while interesting and useful for laboratory practice has one very obvious limitation 
If I have interpreted correctly, Abbot and Roche test recognize IgG for nucleocapsid, TBs instead IgG IgM and IgA for spike and not nucleocapsid. Since if they see different things I don't understand the point of comparison. Or is it me who does not know well the methods, which therefore must be however better clarified both in the mat and met section and in the results? In fact, in figure 1 we talk about generic anti-sars antibodies ... but if they recognize spike and nucleoprotein must be specified in the figure because it is confusing to see the two tests one that sees the spike and the other that sees the nucleoprotein.
In figure 3 the x-axis numbers are not read well, either figure 1c. Figure 5 is unclear and the legend does not help to understand well what is represented in A and what in B.
Therefore, the work requires a thorough and timely revision of the work, which in its current form is not publishable.

Author Response

Dear Reviewer 1,

Many thanks for your kind and useful feedback for our paper.

Please find the below responses and explanation of amendments we have made:

  • You wanted clarification as to why we compared a nucleocapsid and spike assay in our study:
    • Our study has used samples derived from individuals who are convalescent from natural COVID-19 infection, not vaccination.
    • Therefore, as highlighted in "study design" of material and methods section, we have explained, that natural infection elicits a polyclonal response to multiple SARS-CoV-2 structural sites; such as Nucleocapsid and spike, not one or the other.
    • Therefore the choice of antigen used, nucleocapsid or spike, within the immunoassay is not critical to detect natural infection.
    • However, the manner of setting assay thresholds based on validations, which have been conducted only on severe hospitalised COVID-19 patients, can lead to over-exaggerating assay thresholds. This is seen with the Abbott, and we have shown this in our study. Abbott verified their assay primarily on severe COVID-19 patients, with high viral loads. This meant their threshold was set too high, which failed to detect low positive samples; as seen in non-hospitalised convalescent  individuals.
    • This leads to underestimation of seroprevalence by Abbott; which was the main immunoassay used to conduct serological testing in UK following its roll-out.
  • I have changed the x-axis on Figure 1c and 3, so the numbers are a bit clearer.
  • Legend on Figure 5:
    • I have reworded the legend description.
    • Figure 5 aims to show the comparison of serological outcomes from use of the manufacturer Abbott threshold (Abbott 1.40) and our new threshold for Abbott of 0.64.
    • We have applied these threshold and measured the serological outcomes from measuring samples in Portsmouth (Figure 5a) and DGH (Figure 5b).
    • In this manner, we can see how the new Abbott threshold 0.64 compares with the old Abbott threshold 1.4, and from outcomes produced by with TBS (Fig 5a) and Roche (Fig 5b).

Reviewer 2 Report

I read carefully the manuscript "Comparative assessment of SARS-CoV-2 serology in healthcare workers with Abbott Architect, Roche Elecsys and The Binding site ELISA immunoassays" and in my opinion it needs minor revision. It is very interesting and relevant paper. It only requires adding a few details.

I have listed my minor remarks below:

  • Is it possible to simplify the title a bit? Too detail at the very beginning may discourage the potential reader. Maybe "selected assays" instead of specific names?
  • Please, explain all abbreviations used in the abstract (SARS-CoV-2, IgG, PCR).
  • Use keywords different than in the title.
  • Line 53 - explain PCR.
  • Line 58 - explain IgM and IgG.
  • Line 99 - explain US FDA.
  • Please, check missing spaces in the text.
  • Figure 1 - Would changing the graph type make the data more readable?
  • Add a paragraph "Conclusion".

Best regards,

Reviewer

Author Response

Dear Reviewer 2,

Many thanks for your kind and informative feedback. Please find following the amendments we have made to your comments:

  • Change in title:
    • I have changed the title, so its a bit easier to understand and read; "

      Antibody responses to SARS-Cov-2 infection - comparative determination of seroprevalence in two high-throughput assays versus a sensitive spike protein ELISA

  • I have explained the following abrreviations in the Introduction and materials and methods:
    • SARS-CoV-2
    • PCR
    • IgG
    • IgM
    • FDA and MHRA
  • I have added a conclusion, which I've explained the main findings of our study and how it answers our main scientific question; does Abbott assay underestimate COVID-19 seroprevalence, which we have proved in this study.

Round 2

Reviewer 1 Report

THE AUTHORS HAVE RESPONDED ADEQUATELY TO ALL MY REQUESTS AND I CONSIDER THE WORK PUBLISHABLE IN THIS FORM